# Trade liberalization and total factor productivity in Brazil: A vecm modeling

**Edivo Oliveira de Almeida[1], Julio Vicente Cateia[2]\*, William Barbosa[3], Clailton Ataides de Freitas[4]**

1 Institute of Economics, State University of Campinas, Campinas, SP, Brazil, 2 Department of Economics, Federal University of Piaui, Teresina, PI, Brazil, 3 Esalq, University of Sao Paulo, Piracicaba, SP, Brazil, 4 Graduate Program of Economics and Development, Federal University of Santa Maria, Santa Maria, RS, Brazil

\* juliocateia@yahoo.com.br

## Abstract

This paper aims to investigate trade openness's effects on total factor productivity (TFP) using monthly data from December 1991 to March 2024. The analysis also incorporated absorptive capacity to examine the behavior of TFP components. Our findings from a multivariate VECM model indicated that absorptive capacity did not significantly impact TFP, even in the short term. Conversely, the increase in openness contributes to raising TFP by about 26 and 0.16 percentage points in the short and long term, respectively. Additionally, absorptive capacity and trade openness Granger-Cause short- and long-term components of TFP. These results are statistically significant at conventional significance levels. Policymakers should consider the dynamic effects of their policy actions on other sectors of the economy that were not initially the focus of the policy. Policymakers should develop concrete policies that improve the efficient use of resources in production chains to potentialize the productive impact of trade liberalization, including investment in human capital, ICT, and research and development.

## 1. Introduction

The process of greater international integration of the Brazilian economy took place in the early 1990s with the trade liberalization policy, when import tariffs were reduced for almost all products [1], 2006), but mainly with the creation of the Southern Common Market (Mercosur) in 1991, to boost regional trade. Thus, after decades of industrial protection and periods of high inflation rates, especially in the 1980s, throughout the 1990s, Brazil underwent a necessary structural transformation because of economic opening and privatization associated with stability, which resulted in solid incentives for national and foreign investments, advancing industrial restructuring in the country, although more quickly in some sectors than in others [2]. Economic openness had as an immediate consequence a significant increase in foreign direct investment (FDI) flows, strongly associated with rationalization and modernization of the productive structure underway after implementing the *Plano Real*, an exchange rate-anchored monetary policy reform in the early 1990s [3].

**Data Availability Statement:** All relevant data are included within the article and its supporting information files. The data used to estimate the model are publicly available online and can be

accessed via the following link: http://www.
ipeadata.gov.br/Default.aspx.

**Funding:** The author(s) received no specific
funding for this work.

**Competing interests:** The authors have declared
that no competing interests exist.

The debate on Brazilian economic development after structural reforms has been centered on the implications that such reforms could bring to the industrial restructuring process. Scholars such as Barro and Goldenstein [4] argue that trade has advanced the economic integration of the Brazilian economy since the import increase was associated with the demand for components and machines by companies located in the country as a way of production cost reduction and modernization to face increased competition. On the other hand, several authors, such as Coutinho [5], defend the idea that economic opening led to a regressive specialization of the Brazilian industrial sector since the restructuring of the industry occurred towards natural resources-intensive sectors and with low value-added rather than technology-intensive sectors. Therefore, the structural reforms of the 1990s would harm development because they would further distance Brazil from developed nations by not allowing the creation of endogenous capacity for technological innovation.

Various empirical studies have examined the economic impacts of changes in the Brazilian economy after trade liberalization. These studies have looked at the effects of tariff reductions on factor reallocation and productivity [6–8], job flows between sectors, regional migration, and wage structure [9,10]), as well as the welfare implications of these changes [11,12]. Additionally, researchers have assessed Brazil's economic integration in terms of interregional FDI allocation and its influence on growth [13,14].

However, empirical studies on the relationship between economic openness and total factor productivity (TFP) are still scarce in Brazil. But one of the most important focuses of analysis in recent decades, in the context of theoretical and empirical economic research, is identifying the cyclical and permanent components of financial variables. Existing studies have investigated whether short-term shocks and fluctuations can influence the long-term behavior of variables and, if so, to what extent. There are two ways to understand this question. First, transitory changes do not significantly affect the long-term growth trend of macroeconomic variables such as GDP and employment. Thus, the movement verified will be less abrupt, and short-term macroeconomic fluctuations will fundamentally be explained by changes on the demand side [15]. Conversely, the short-term cyclical fluctuations may be assumed to represent a large part of the long-term trajectory in economic variables. This second hypothesis occurred after the second generation of business cycle literature. According to this literature [16], time series commonly have a unit root or a persistent stochastic trend. The work by Beveridge and Nelson [17] theoretically contributed to past studies by separating dynamics components into a cyclical stationary element with zero means from the persistent stochastic trend. Following their methodology, Cambell and Mankiw [18] demonstrated for selected developed countries a strong persistence of real output over time and a process of gradual reversal of shocks in the US economy. In addition, they observed that short-term shocks tend to be fully absorbed within approximately ten years. On the other hand, by using the Solow Model for sixteen OECD economies, Michelacci and Zaffaron [19] concluded that the per capita incomes of these countries are stationary in the short term but reversible in the medium and long term, indicating that the process can be classified as fractional unit root. Thus, the extended memory property of the series reveals that the convergence process can happen, however, subject to a low speed.

For the Brazilian economy, Cribari Neto [20] using unit root tests and persistence analysis, applied the Beveridge-Nelson decomposition for the GDP of Brazil and Colombia between 1950 and 1985. They showed that innovations in both countries are more persistent than a casual stroll and that stabilization policy has long-term effects. Despite not using the Beveridge and Nelson decomposition method, it is worth emphasizing other works on the relationship between growth and productivity in the Brazilian economy. For instance, Gomes et al. [21] associated the slowdown in the GDP growth rate in Brazil with the fall in TFP between the

mid-1970s and early 1990s. Bacha and Bonelli [22] revealed that one of the main factors that explained the upward trajectory of the Brazilian GDP between 1974 and 1984 was the 2.6% increase in capital stock, which allowed the maintenance of the average GDP growth rate at 3.9% in the same period.

We build on and advance past studies in several directions. First, several recent studies suggest a relationship between productivity and trade openness. For example, Fereira and Cateia [23] assessed the implications of trade openness on productivity and its effects on structural change in Guinea-Bissau. They found that trade was responsible for generating the labor economy in agriculture. These workers migrated to manufacturing and service sectors that were enjoying positive investment externalities provided by the gains from trade after the openness. However, their study does not decompose post-trade liberalization productivity. This paper follows several developments in real business cycle theory [24], decomposing TFP into permanent and cyclical components. The cyclical component shows the intertemporal equilibrium of macroeconomics, while the cyclical component reflects some agents' decisions that affect, for example, the supply of productive factors after opening. Thus, this treatment will help us identify the real causes of the opening of TFP in Brazil.

Second, we apply the vector error correction model (VECM) approach to analyze the contemporary and past effects of a shock to one variable on the other variables in the model. Thus, the VECM model allows us to verify the permanent and cyclical components of the TFP.

Finally, Fig 1 shows the behavior of TFP and openness, which is the sum of imports and exports normalized by GDP. This is the critical variable of the model. Still, given the VECM, we can also analyze the effect of productivity on other variables, such as absorptive capacity, trade openness, Etc.

The remainder of this paper is structured as follows. Section 1 reviews the real business cycles literature. Section 2 outlines the Brazilian economic dynamics from the second half of the last century to the recent period. Section 3 describes the empirical model. Section 4 presents the results. Section 6 concludes.

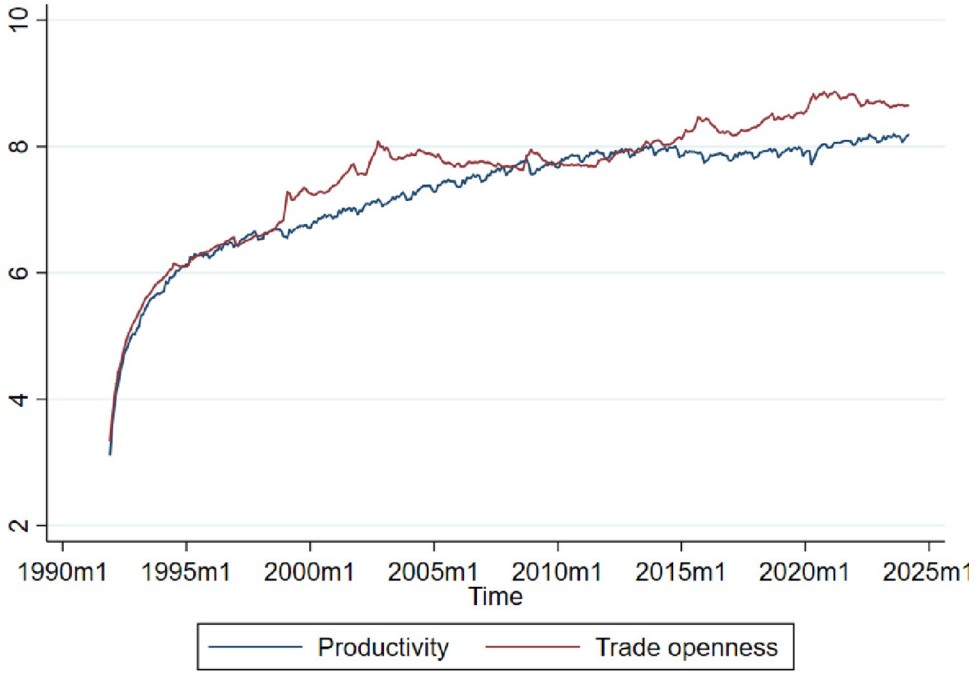

**Fig 1. Productivity and trade openness.** Source: The authors.

## 2. Literature review and Brazilian specificities

Although studies of the gains from trade date back to the early work by classical economists, it was after World War I that it became a systematic research agenda for economists. Samuelson [25] was one of the precursors of this literature, and Krugman, in several papers in the 1980s, studied gains from trade for various competition settings. In Samuelson's own words, it isn't easy to demonstrate rigorously that trade openness is better for all countries. Still, it is possible to conclude that an open economy is preferable to a closed economy. However, explaining why trade openness is desirable over a closed economy involves precisely explaining the sources of gains from openness unavailable under autarky. Trade can be on a continuum of goods [26–28], investment and the production content it brings in terms of FDI or multinational production [29–32].

Economics can be more varied and spread differently across countries, sectors of activity, and classes of society. For developed countries, trade helps to expand the size of these countries' markets, spread technology, and import inputs they need to expand local production, which can improve their current growth rate [33]. For developing countries, trade allows countries to gain access to manufactured goods that they lack and access to technology. They will also be able to export their agricultural products, obtaining gains from trade and reinvesting them in expanding national productive capacity or diversifying the economy.

Krugman [34] develops a model of trade that demonstrates that trade leads to intra-industry specialization, which reflects scale economies and consumers' taste for a diversity of products. Trade can also increase the variety of goods available. This stream of love-of-variety models argues that increasing product variety is the leading cause of gains from trade. Some recent studies also demonstrate an income effect associated with variety [35–37]. For Betts et al. [38], trade-in variety can promote structural change.

Building on Krugman [39], some studies develop models with increasing returns technology under imperfect competition. One of the most recent studies was conducted by Kokovin et al. [40], which demonstrated welfare gains when the economy is far from the risk of autarchy. Fluctuating markups and underpriced imports cause fluctuating societal consequences, which harm small-scale trade.

The Ricardian comparative advantage models provided a unified approach to innovation, growth, and trade in the work by Eaton and Kortum [26]. Expanded trade corresponds to lowering geographical barriers, and productivity research facilitated its positive implications for economic growth. Trade openness used in most empirical studies is a quotient calculated as imports plus exports over nominal GDP. The size of the index indicates how open an economy is in each period. However, Alcalá and Ciccone [41] proposed an alternative measure based on real purchasing power parity. Their openness index is defined as imports plus exports in exchange rate relative to GDP in purchasing power parity. They call this new index real openness, in contrast to nominal openness. The detail of why real openness is preferred has been discussed by them and several subsequent studies. In our case, it allows us to relate real openness to the characteristics of the Brazilian labor market, which may also influence factor productivity.

Männasoo et al. [42] investigated the factors influencing total factor productivity (TFP) growth in 99 European regions across 31 countries from 2000 to 2013. They found evidence that higher levels of human capital positively impacted TFP growth, especially in the advanced areas. In contrast, the impact of regions' own research and development (R&D) expenditures on TFP growth was mainly small. Our study also incorporates R&D into the analysis of the determinants of productivity. In contrast, the VAR structure we adopted allows us to analyze the feedback and time effects of shocks to the model variables. Another study we also built on

is that of Nakamura et al. [43] for Japan. They found that the decline in productivity in Japan in recent decades is due to the poor management of accumulated ideas. Thus, improved productivity would be achieved by encouraging the flexible reallocation of management resources at the corporate level, such as capital and labor.

Guan and Cheng [44] use firm-level data from China from 2000 to 2006 to examine the product complexity-productivity linkage at the firm level. Trade is reported to have increased product complexity, which positively affected productivity. The effects vary across different sectors and provinces; technological factors explain the impact of complexity on productivity in both settings. Yu et al. [45] studied the effects of trade liberalization on firm productivity. They found that trade increases the productivity of firms producing complex products but not the productivity of firms producing simple products.

For Brazil, Versiani and Suzigan [46] explain that the crisis in the Brazilian primary-exporting sector and the subsequent actions of the State to protect the coffee sector were indirectly crucial for the development of the national industry. Thus, the beginning of industrialization in Brazil increased demand for manufactured products due to the accumulated internal income in export activities, allied to protectionist policies that took the form of exchange rate devaluation, direct control over the exchange market, or quantitative control over imports. However, they argued that it was only from the second half of the 1950s onwards that policymakers could identify a well-defined strategy explicitly aiming at the modernization of the economy.

The *Plano de Metas* (target plan-PM) was an investment program aimed at expanding the capacity of infrastructure supply in various dimensions in the mid-1950s. According to Arend [47], with the PM, Brazil sought to internationalize its economy, attracting foreign companies and promoting changes in its regulatory framework to facilitate the penetration of foreign capital. However, with the advent of the Military Regime in the 1960s, Brazil experienced an ambiguous scenario, marked by complex internal contradictions configured in an economy that was becoming one of the largest in the world but at the expense of high-income inequality.

The boom in the Brazilian economy was between 1968 and 1973, known as the "Brazilian Miracle," in which the country sustained robust growth and higher investment rates, primarily provided by the international liquidity scenario, as well as by the expansion of the world economy and for the institutional reforms promoted by the Economic Action Program (PAEG). This favorable scenario suffered a strong reversal with the two oil crises in 1973 and 1979. These adverse shocks reduced liquidity while increasing international interest rates to the detriment of external stimuli, essential for domestic growth. In short, from the 1950s to the 1970s, Brazil experimented with a series of programs and plans whose massive investments were mainly directed to the industrial sector, increased industrial production, and introduced the country to the list of emerging countries undergoing industrialization. Fig 2 shows Brazil's gross fixed capital formation (GFCF) between 1950 and 2008. Throughout this period, it began to show a persistent upward trend until the first oil shock. From 1974 to 1980, the variation in GFCF decreased but remained at high levels.

The 1980s can be characterized as a period of solid stagnation of GDP per capita, increased income inequality, and uncontrolled acceleration of the inflationary process [48]. There were considerable repercussions on the industry's performance, with most of the decade marked by acute macroeconomic and monetary distortions, which caused a reversal in the behavior of investments in the Brazilian economy. In Figure, it is possible to verify that from the first years of the 1980s onwards, the GFCF variation remains below the average for the entire period. That denotes the unfavorable expectations regarding the expected investment returns during this decade.

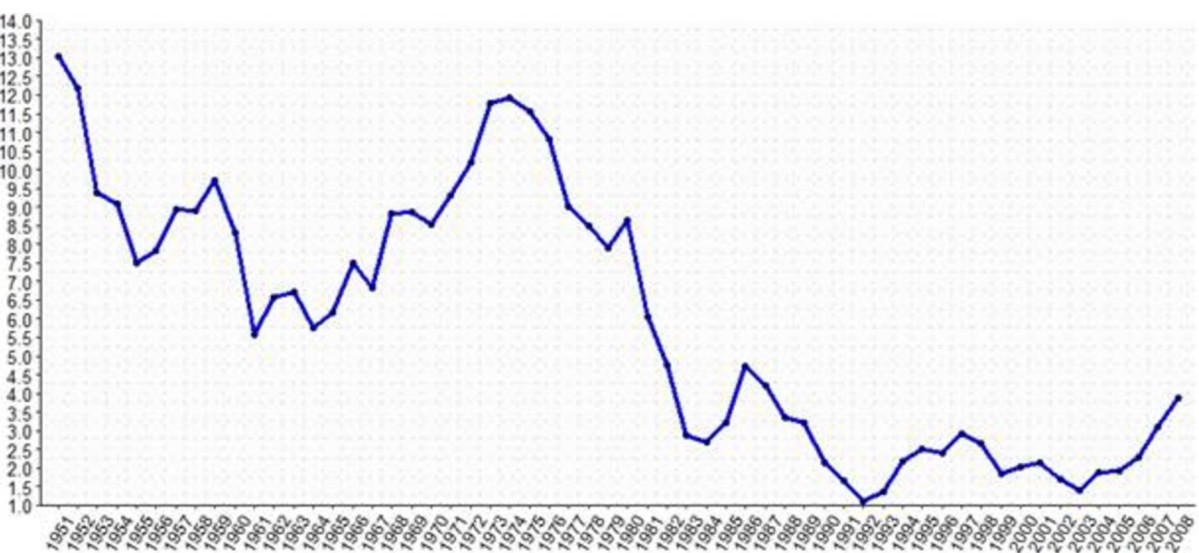

**Fig 2. Gross fixed capital formation in Brazil between 1950 and 2008.** Source: The authors. IPEADATA database.

After various monetary stabilization plans, the Plano Real, launched in 1994, successfully achieved this goal. In fact, since 1992, it is possible to verify that the GFCF variation began to show advances relative to previous years but became more volatile in the period that coincided with the financial crises at the end of that decade. Nevertheless, the net public debt increases and the indiscriminate economic opening, among other measures, have been listed as a negative point inherited by Brazil from 1994–2001 [49]. Between 2002 and 2010, the Brazilian economy was marked by growth, price stability, and income transfer programs to eradicate poverty. However, it was also characterized by the deterioration of the current account balance and the export agenda based on raw materials [50]. The GFCF variation also remains close to that verified in the second half of the 1990s. However, as of 2006, the GFCF behavior was significantly more favorable than in previous years.

These assertions can be seen in Fig 1 of the previous section through the nuances and behavior presented by the effective labor productivity in Brazil in recent periods. The GFCF does not coincide precisely with that of labor productivity, but both variables are dynamically related to each other. Labor productivity has expanded continuously between 2000 and 2010. The reversal of the dynamics experienced by the Brazilian economy from the 2010s onwards coincides with the negative behavior that investments presented from that decade forward. It is worth noting that, on average, this productivity at the end of the 1980s was critically lower than that observed in the early years of that same decade. Some improvement in productivity performance is identified from 1992 onwards. However, productivity lost steam from 1998 onwards, potentially due to the Russian financial crisis, which intensified the volatility of investments in Brazil during that period. Until the mid-2000s, the unsatisfactory behavior of labor productivity outcomes was maintained despite some recovery from 2006 onwards, reflecting the strong impulse caused by the GFCF variation. In 2008, it was possible to verify that Brazilian productivity remained at levels like those in early 1980. The threshold years, namely 1980 to 2010, reflect about three decades of almost stagnation of effective labor productivity in Brazil.

Based on this macro scenario, we subsequently establish the theoretical framework using the Beveridge-Nelson [17] decomposition method, the VAR model, and the Granger Causality test.

## 3. Methodology

This section presents the VAR model that will be used to investigate how a shock to one variable impacts the other variables in the model and itself in a contemporaneous and lagged manner. After presenting the VAR model, we show the strategy for decomposing productivity. The section concludes by presenting a list of the variables and the data sources used to estimate them.

### 3.1 Vector Autoregressive (VAR) models

Before DSGE settings, VAR modeling was the most critical development in macroeconometrics in the 1980s. The Box-Jenkins methodology had the following two main limitations: (i) the study of a single variable is not practical, and it may be hard to derive from it a credible policy recommendation, and (ii) time series present fluctuations that are generally interrelated and persistent. Therefore, a forecast taking both short- and long-term dynamics provides a flexible and tractable picture of covariations over time and politically significant results [51,52].

A convenient way to model shocks in the economy is through a system of equations where the impact of a shock on one variable is shared across the closed system. More generally, a structural VAR model can be specified as follows:

$$AX_t = B_0 + B_1X_{t-1} + B_2X_{t-2} + \cdots + B_iX_{t-i} + B\varepsilon_t \tag{1}$$

where $A$ is a matrix of contemporary restrictions that indirectly captures interrelations between shocks in the system; $X_t = (X_{1t}, X_{2t} \ldots, X_{it})$ is a vector containing macroeconomic series. According to Eq (1), these random variables are endogenously specified; $B_0$ is a $nx1$ vector of constants; $B_i$ is a $nxn$ matrix of coefficients; $B$ is a diagonal matrix of deviations of order $nxn$; and $\varepsilon_t = (\varepsilon_{1t}, \varepsilon_{2t} \ldots, \varepsilon_t)$ is an $nx1$-dimensional vector of independent and identically distributed unobserved random errors with zero mean and constant variance, that is, $\varepsilon_t \sim i.i.d(0, \rho)$, where $\rho$ is a positive definite matrix that contains variance and covariance structures.

An empirical challenge underlying system (1) is identification because the coefficients appear on both sides of the system. Thus, to perform the estimation, a more convenient way consists in pre-multiplying the system by the inverse of the contemporary constraint matrix $A^{-1}$ [53]. By doing this, we obtain the reduced form:

$$X_t = A^{-1}B_0 + \sum_{i=1}^{p} A^{-1}B_iX_{t-i} + A^{-1}B\varepsilon_t \tag{2}$$

Let $\phi = A^{-1}B_i$ and $B\varepsilon_t = Ae_t$. By the invertible matrix property and based on the Granger representation theorem [54], system (2) can be rewritten more generally as:

$$\Delta X_t = \phi X_{t-1} + \sum_{i=1}^{p-1} \wedge_i \Delta X_{t-1} + e_t \tag{3}$$

where $\Delta X_t = (X_t - X_{t-1})$ and $\wedge_i = -\sum_{j=1+i}^{p-1} \phi_j, i = 1, 2, \ldots, p-1$.

System (3) is the vector error correction model (VECM). Several methods can be used for forecasting [55,56]. However, the VECM approach fits well with the purpose of our analysis because it allows us to analyze the short- and long-term effects of shocks. The main advantage of VECM over other existing macroeconometrics models is that it allows us to explain $\Delta X_t$ through short-term factors associated with the second expression on the right-hand side, $\wedge_i$, and the long-term relationship captured by the first expression of Eq (3), $\phi X_{t-1}$. Theoretically, $\phi X_{t-1}$ explains $\Delta X_t$ if both have a common long-term relationship, that is, they are cointegrated. In this case, $\phi(I) = 0$, so that $\phi = \beta\alpha'$, where $\beta$ is a matrix of r cointegration vectors, and $\alpha$ is a matrix of r adjustment vectors [57].

To estimate model (3), we will first specify the variables' lag order using common selection criteria, such as the Akaike information criterion (AIC), the Bayesian information criterion (BIC), and the Hannan-Quinn information criterion (HQ). Then, we apply the Kwiatkowski, Phillips, Schmidt, and Shin (KPSS) test to verify the series' behavior, that is, whether it is stationary. Stationarity is a necessary condition for estimating time series. A time series is considered stationary if its mean and variance are constant and if the covariance depends only on the time lag and not on the period in which it is calculated. If two series have the same order of integration, then they are cointegrated. Engle and Granger [54] present this argument more formally. Suppose each time series vector $z_t$ element is stationary at the first difference and a linear combination $a'z_t$ is stationary. In that case, the time series are said to be cointegrated, with $\alpha$ being the cointegration vector.

## 3.2 Empirical strategy and data

The data for this research are from the Institute of Applied Economic Research (*Instituto de Pesquisa Econômica Aplicada*-Ipeadata). These are monthly data from December 1991 to March 2024, the latest period for which data are available for all variables in the model. The variables are as follows: total factor productivity, trade openness ($x_t$) and gross fixed capital formation ($z_t$) proxying absorptive capacity. Existing literature shows that trade openness and absorptive capacity are one of the main determinants of TFP [58,59].

We decompose the TFP as in Beveridge and Nelson [17] theorem. We assume that time series can be modeled as an ARIMA (p,d,q), generically represented as follows:

$$y_t = \delta + \Theta_1 y_{t-1} + \cdots + \Theta_p y_{t-p} + \Phi_1 \epsilon_{t-1} + \cdots + \Phi_q \Phi \epsilon_{t-p} \tag{4}$$

where $y_t$ is the TFP, of order (p); $\epsilon_t$ is the contemporary white noise error of order (q). Since $y_t$ depends on the contemporary error and the immediately past error, the model can be operationalized using a lag operator as follows:

$$\psi(L) \equiv \frac{\Theta(L)}{\Phi(L)} \tag{5}$$

It is important to emphasize that this model, in level, presents a stochastic tendency of the random walk type. According to Beveridge and Nelson [17], this model is based on the following mathematical identity:

$$\psi(L) \equiv \psi(1) + (1 - L)\psi(L) \tag{6}$$

where $\psi$ is a lag operator ($\psi(L) = \sum_{j=0}^{\infty} \psi_j$) following a random walk with drift. From these identities, $y_t$ can be partitioned into permanent and cyclical components, respectively, $p_t$ and $c_t$, so that:

$$y_t = p_t + c_t \tag{7}$$

$$c_t = \psi(L)\epsilon_t \tag{8}$$

$$p_t = y_0 + \delta_t + \psi(1) \sum_{j=1}^{t} \epsilon_j \tag{9}$$

Therefore, the variable $p_t$ corresponds to the permanent factor of the random walk with drift and innovation with $\psi(1)$ σ, while the $c_t$ consists of the transitory factor of the model. Beveridge and Nelson [17] explain that in the case of a deterministic trend, temporary shocks are attenuated over time, because there is always a well-defined trend line, so that they do not repeat in causal-type stochastic trends.

Thus, the Eq (9) of the permanent component of the model is obtained, which results from the long-term forecast of the series adjusted in relation to the deterministic trend. This is a process that can be inexorably characterized as a random walk. Thus, the cyclic component can be found simply by deducting $p_t$ from $y_t$. In short, Eq (4) can be represented alternatively using the polynomial lag operators such as:

$$\Phi(L)(\Delta y_t - c) = \Theta(L)\epsilon_t \tag{10}$$

where $\Phi(L)$ and $\Theta(L)$ are lag polynomials, so that:

$$\Delta y_t = \delta + \psi(L)\epsilon_t \tag{11}$$

The cyclical component (also called the short term) is obtained by deducting the permanent factor from the original series, $c_t = y_t - p_t$. Thus, the following deductions are made:

$$y_t = (1 + \Phi)y_{t-1} + \delta + \epsilon_t \tag{12}$$

By finding the value of $\Phi$ in Eq (12) and replacing it in Eq (9) we obtain the permanent component of $p_t$ given by:

$$p_t = p_{t-1} + \delta + \left[\frac{1}{(1 - \Phi)}\right]\epsilon_t \tag{13}$$

## 4. Results and discussions

This section presents and discusses the results of the model. It starts with a diagnostic test to verify the stationarity of the time series, a cointegration test, and a selection of the VECM order using information criteria. The null of the KPSS test is that the trend is stationary against the false that it is non-stationary. Table 1 shows that all variables are not zero-order integrated, I(0), but first-order integrated, I(1) because the test statistics are statistically significant at 5%.

Meanwhile, Table 2 reports the results of the AIC, HQ, and BIC information criteria. The HQ and BIC criteria present a significance of 10% in the same lag order. On the other hand, the AIC criterion presents significance only in the fourth lag. According to Cateia et al. [60], the BIC criterion penalizes more than the other information criteria. We can conclude that VECM has a maximum lag order of three in this case. We adopted a parsimonious specification with VECM(2).

Since the series are non-stationary but I(1), we must analyze whether they have a long-term relationship. There are several ways to perform systematic long-term analysis of time series. A more effective way to analyze the behavior of the I(1) series is through the Johansen/trace cointegration test. The Johansen test allows identifying the existence and vectors of the cointegration of the model. We perform this test with a VECM (4), a model with a maximum number

**Table 1. Unit root test.**

| Time series | Lag order I(0) | | Lag order I(1) | |
| --- | --- | --- | --- | --- |
| | Test statistics | Critical value at 5% | Test statistics | Critical value at 5% |
| $p_t$ | 2.720 | 0.146** | 0.048 | 0.146** |
| $c_t$ | 8.310 | 0.146** | 0.038 | 0.146** |
| $z_t$ | 4.100 | 0.146** | 0.080 | 0.146** |
| $x_t$ | 4.660 | 0.146** | 0.050 | 0.146** |

Source: The authors

** $p < 0,05$.

**Table 2. VECM lag order selection criteria.**

| Lag | LL | LR | P | AIC | HQ | BIC |
|---|---|---|---|---|---|---|
| 0 | -13725.3 | | 0.000 | 71.5066 | 71.5229 | 71.5477 |
| 1 | -10110.3 | 7229.9 | 0.000 | 52.7621 | 52.8437 | 52.9678 |
| 2 | -9960.29 | 300.05 | 0.000 | 52.064 | 52.2109* | 52.4344* |
| 3 | -9933.87 | 52.847 | 0.000 | 52.0097 | 52.2219 | 52.5447 |
| 4 | -9916.73 | 34.282* | 0.000 | 52.0038* | 52.2813 | 52.7034 |

Source: The authors

* $p < 0.1$. Note: LL = Likelihood-ratio statistic (for formal definition, see Hamilton [53].

of four lags and a constant trend (Table 3). Starting from a maximum rank of zero, we do not reject the null hypothesis that there is no cointegration. Conversely, at the maximum rank of 1, we reject the null at a significance level of 10%, suggesting that there is at least one cointegration vector.

We proceeded with its estimation once the model successfully passed the relevant evaluation and diagnostic tests. Thus, we divided the VECM(2) model's results into two parts: the short-term effects (Table 4) and the long-term impacts (Table 5). As a consistency analysis, we estimated a VAR(2) model and then performed the Granger causality test (Table 6).

The short-term coefficients reported in Table 4 allow us to assess the effects of a shock to a variable on the other variables in the model and its feedback. However, we focus on the policy equation, that is, on studying the effects of absorptive capacity ($Z_t$) and trade openness ($X_t$) on the permanent ($P_t$) and cyclical ($C_t$) components of TFP. All else being equal, at the business-as-usual level, a 1% shock in absorptive capacity causes a decrease of about 0.52 percentage points in the permanent component of TFP. This effect is statistically significant at conventional significance levels. Similarly, a 1% increase in absorptive capacity contributes to reducing cyclic TFP by 0.12 percentage points, but this result is now statistically insignificant. On the other hand, absorptive capacity has a positive and statistically significant contemporaneous effect on itself and a significant positive impact of about 0.0025 percentage points on trade results, possibly due to its effect on GDP. Increasing absorptive capacity raises economic activity and can increase exports, contributing to raising GDP.

The negative outcomes of absorptive capacity on productivity were not expected because productivity theoretically increases with increasing absorptive capacity. However, it is a result consistent with the Brazilian reality, at least during the period analyzed in this study. Some past studies [61–64] suggest that factors such as infrastructure, ICT, and the tax system may be critical determinants of factor productivity in developing countries with agriculture-based

**Table 3. Johansen /Trace cointegration test.**

| Ranking max. | Parameters | Eigenvalues | Trace |
|---|---|---|---|
| 0 | 52 | 0.000 | 83.661 |
| 1 | 59 | 0.149 | 21.614* |
| 2 | 64 | 0.041 | 5.482 |
| 3 | 67 | 0.0141 | 0.0291 |
| 4 | 68 | 0.00008 | 0.000 |

Source: The authors

* $p < 0.1$.

**Table 4. Short-term coefficients.**

| Time series | $P_t$ | $C_t$ | $Z_t$ | $X_t$ |
|---|---|---|---|---|
| $P_t$ | -0.012 (0.051) | -0.123 (0.043)*** | -0.056 (0.014)*** | -0.0004 (0.0002) |
| $C_t$ | 0.063 (0.049)*** | 0.596 (0.041)*** | 0.007 (0.014) | -0.0002 (0.0002) |
| $Z_t$ | -0.526 (0.180)*** | -0.122 (0.152) | 0.193 (0.051)*** | 0.0025 (0.0009)*** |
| $X_t$ | 26.282 (8.732)*** | 9.175 (7.391) | -1.987 (2.487) | -0.460 (0.046)*** |

Source: The authors

*** p < 0.01; () standard deviation.

economies. This would be justified because better infrastructure quality and access to ICT reduce transaction costs, and a tax system schematically designed to collect and allocate resources for productive purposes helps to spread productivity externalities. Scholars argue that transaction costs in Brazil are high and have referred to them as Brazil Costs since the opening of trade in the early 1990s [65]. A single factor does not explain Brazil's cost; however, the main determinants are the low supply and quality of physical and technological infrastructure and high customs burdens.

These factors tend to inhibit some sectors' productivity gains in recent decades. For example, Bustos et al. [66] show that technological innovation in agriculture, such as the development of genetically improved seeds by the Brazilian Agricultural Research Corporation (*A Empresa Brasileira de Pesquisa Agropecuária*-Embrapa), has increased agricultural productivity. Improved productivity could accelerate structural change in the country. However, the authors emphasize an exciting aspect that could hinder this process: regional development disparities since some Brazilian states are still extremely poor and have low productivity. Examples are the northern and northeastern states, whose human and sectoral development levels are lower than the national average. Therefore, although agricultural productivity has increased due to innovations in agriculture, most of the productivity gains are concentrated in Brazil's southern and southwestern regions and, to a lesser extent, in the central-western region.

Meanwhile, we observed that trade openness was the primary determinant of TFP. An increase of 1 in the degree of trade openness contributes to an increase in permanent TFP by

**Table 5. Cointegration and adjustment vectors.**

| Time serie | Long-term adjustment vector | At least one vector of cointegration |
|---|---|---|
| $p_t$ | 0.0019 (0.0008)** | 1 |
| $c_t$ | 0.006 (0.0007)*** | -0.093 (0.106) |
| $z_t$ | 0.0008 (0.002) | 1.854 (0.457)*** |
| $x_t$ | 0.163 (0.138) | 0.038 (0.018)* |

Source: The authors

*** p < 0.01

* p < 0.1.

about 26.28 percentage points and cyclical TFP by 9.17%. The impact of openness on productivity is significant, even at 1%, but statistically insignificant in explaining the cyclical component of TFP.

Our analysis also examined the long-term impact of trade openness and absorptive capacity on total factor productivity (TFP) (refer to Table 5, Column 2). Our findings indicate that long-term changes in productivity, both permanent and cyclical, affect the long-term behavior of macroeconomic variables. Therefore, the policy implications are that policies aimed at boosting productivity can enhance absorptive capacity and amplify the benefits of economic openness on factor productivity. The results reveal a connection between productive sectors and the external sector. Specifically, we observed that the productivity components are higher than their equilibrium value (Column 3), suggesting that if TFP is exceptionally high and exceeds the level sustainable by factor employment, it could increase absorptive capacity and enhance the economy's degree of openness.

Finally, we estimate the VAR(2) model and then perform the Granger causality test. According to Song and Taamouti [67], the Granger causality test is one of the most reliable macroeconometrics tests for studying the causal relationship between variables (Table 6).

Using the Granger causality test, we find that TFP, absorptive capacity, and trade openness Granger impact the other variables in the model as in Cateia and Savard [68]. This effect is significant at the 1% level. These results suggest that policymakers who, for example, wish to increase productivity through trade openness should also take into account the effects of this policy on other macro variables, such as the level of economic activity or absorption. Notably, the $X_t$ and $Z_t$ Granger-Cause the short-and long-term components of labor productivity in Brazil. This result implies that an increase in openness raises short-and long-term productivity. Thus, changes in the short-term performance of labor productivity were preceded by changes in the same direction in the variation of investments.

Existing studies by Solow [69] demonstrated the intrinsic relationship between capital investments and factor productivity. Bacha and Bonelli [22] showed the close relationship between the physical capital accumulation rate and the performance of the Brazilian economy. Since in the previous sections, a decline in the effective productivity $y_t$ and of the short-term component ($C_t$) was identified, from the late 1970s and early 1980s onwards, the application of the test of Granger causality between the investments and the series considered from that decade on, became imperative for the proper understanding of this phenomenon.

Overall, the results indicate that the short-term trend in productivity depends to some extent on absorptive capacity and trade liberalization. However, as the long-term element has

**Table 6. Granger causality Wald tests.**

| H0 | Chi2-test | Prob > chi2 |
|---|---|---|
| $p_t$ Granger-Causes all | 24.663*** | 0.000 |
| $c_t$ Granger-Causes all | 23.941*** | 0.001 |
| $z_t$ Granger-Causes all | 22.408*** | 0.001 |
| $x_t$ Granger-Causes all | 22.493*** | 0.001 |
| $z_t$ Granger-Causes $p_t$ | 14.153*** | 0.001 |
| $x_t$ Granger-Causes $p_t$ | 18.817*** | 0.000 |
| $z_t$ Granger-Causes $c_t$ | 1.508 | 0.470 |
| $x_t$ Granger-Causes $c_t$ | 6.087* | 0.048 |

Source: The authors

*** p < 0.01

* p < 0.1.

a persistent trajectory and does not revert to short-term shocks in productivity, it is inferred that heavy investments up to the 1970s engendered the persistence of the long-term course, even after the unfavorable reversal investments after 1980. In addition, the existence of Granger Causality between the short and long term reinforces the inference obtained previously in which the presence of a unit root in the effective TFP, exploratory signals a persistent character of the component of the term that does not reverse to short-term shocks.

Therefore, the short-term factor remained above the long-term trend until part of the 1980s, due to the investment packages established up to that period, as shown in Fig 1; and that by the Granger Causality test, it is possible to state that they precede the behavior of productivity in the short run. With the fall in investment packages, the long-term trend—which does not revert to short-term shocks—continued to rise and outpaced the short-term factor.

With the individual analysis of the short and long-term components, it was possible to diagnose how stagnant the Brazilian labor productivity was in the evaluated period. That is because the effective productivity in 2010 did not differ significantly from the levels observed in the early 1980s. On the other hand, its short-term component in 2010 was at levels like those observed in the early 1950s. This inference will require further studies and deepening in the identification of the factors that led to such behavior, that is, the phenomenon designated in the scope of this work as the enigma of labor productivity in Brazil, but which, as a hypothesis, it is suggested is associated the absence of integrated and systematic Economic Plans for development in more dynamic industrial sectors.

## 5. Conclusions

The main objective of this paper was to evaluate the impact of trade openness on total factor productivity (TFP) using monthly data from December 1991 to 2024. It also includes absorptive capacity to investigate the behavior of TFP components. Through a multivariate VECM model, we found evidence that absorptive capacity had no significant impact on TFP, even in the short term. Conversely, the increase in openness contributes to raising TFP even in the long term. This result is statistically significant at conventional significance levels. We also demonstrate the robustness of this result by applying the Granger causality test. In this case, absorptive capacity and trade openness Granger cause an increase in productivity. There is a long-term relationship between these variables, as demonstrated by the cointegration test.

The political implications of these results are enormous. Economic policymakers should take into account the dynamic effects of their policy actions on other sectors of the economy that were not initially the focus of the policy. Economic openness increases production and improves the country's absorptive capacity over time. As a result, although absorptive capacity alone cannot increase TFP in the short term, the impact of openness on it has had positive long-term effects on TFP. In fact, the insignificant short-term results can be attributed to the high transaction costs in Brazil, which result from the low capacity to supply quality infrastructure in several dimensions. Thus, policymakers should develop concrete policies to increase the infrastructure supply and reduce private investment costs. In addition, policies that improve the efficient use of resources in the production chains of the agricultural and industrial sectors should be encouraged. Some concrete measures that can enhance the effect of openness on TFP include investment in human capital, ICT, and research and development.

## Supporting information

**S1 Data.**
(XLSX)

## Author Contributions

**Conceptualization:** Edivo Oliveira de Almeida, Julio Vicente Cateia, William Barbosa, Clailton Ataides de Freitas.

**Data curation:** Edivo Oliveira de Almeida, Julio Vicente Cateia, William Barbosa, Clailton Ataides de Freitas.

**Formal analysis:** Edivo Oliveira de Almeida, Julio Vicente Cateia, William Barbosa, Clailton Ataides de Freitas.

**Funding acquisition:** William Barbosa.

**Investigation:** Edivo Oliveira de Almeida.

**Methodology:** Edivo Oliveira de Almeida, Julio Vicente Cateia, William Barbosa, Clailton Ataides de Freitas.

**Supervision:** William Barbosa.

**Validation:** Julio Vicente Cateia, Clailton Ataides de Freitas.

**Visualization:** Julio Vicente Cateia, Clailton Ataides de Freitas.

**Writing – original draft:** Edivo Oliveira de Almeida, William Barbosa, Clailton Ataides de Freitas.

**Writing – review & editing:** Edivo Oliveira de Almeida, Julio Vicente Cateia, William Barbosa, Clailton Ataides de Freitas.

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
