## [Decision Letter · Decision Letter 0]

21 Feb 2024

PONE-D-23-38514A time series analysis of labor productivity in Brazil (1950-2010)PLOS ONE

Dear Dr. Cateia,

Thank you for submitting your manuscript to PLOS ONE. After careful consideration, we feel that it has merit but does not fully meet PLOS ONE’s publication criteria as it currently stands. Therefore, we invite you to submit a revised version of the manuscript that addresses the points raised during the review process.

**After reading carefully the paper, I fully agree with the reviewers' comments regarding the novelty of the study. Why do you think this study is relevant compared to others on the same topic? The data used in the study are too old (150-2010) and all econometric tests should performed. Most of the comments raised by reviewers are common. Please carefully, respond to these comments and improve the quality of the paper.**

We look forward to receiving your revised manuscript.

Kind regards,

Essossinam Ali, Ph.D

Academic Editor

PLOS ONE

Journal Requirements:

Whilst you may use any professional scientific editing service of your choice, PLOS has partnered with both American Journal Experts (AJE) and Editage to provide discounted services to PLOS authors. Both organizations have experience helping authors meet PLOS guidelines and can provide language editing, translation, manuscript formatting, and figure formatting to ensure your manuscript meets our submission guidelines. To take advantage of our partnership with AJE, visit the AJE website (http://aje.com/go/plos) for a 15% discount off AJE services. To take advantage of our partnership with Editage, visit the Editage website (www.editage.com) and enter referral code PLOSEDIT for a 15% discount off Editage services. If the PLOS editorial team finds any language issues in text that either AJE or Editage has edited, the service provider will re-edit the text for free.

3. In the online submission form, you indicated that Data will be shared upon request.

Additional Editor Comments:

After reading carefully the paper, I fully agree with the reviewers' comments regarding the novelty of the study. Why do you think this study is relevant compared to others on the same topic? The data used in the study are too old (150-2010) and all econometric tests should performed. Most of the comments raised by reviewers are common. Please carefully, respond to these comments and improve the quality of the paper.

Reviewers' comments:

Reviewer's Responses to Questions

**Comments to the Author**

1. Is the manuscript technically sound, and do the data support the conclusions?

Reviewer #1: Yes

Reviewer #2: Partly

2. Has the statistical analysis been performed appropriately and rigorously? 

Reviewer #1: No

Reviewer #2: No

3. Have the authors made all data underlying the findings in their manuscript fully available?

Reviewer #1: No

Reviewer #2: No

4. Is the manuscript presented in an intelligible fashion and written in standard English?

Reviewer #1: Yes

Reviewer #2: Yes

5. Review Comments to the Author

Reviewer #1: 1. Title: The title of this study is general and should be more specific. The study period should not be mentioned in the title.

2. Abstract: (1). The results of the study are in a qualitative way. It is suggested to report here the findings of the study in a quantitative way (2). It is suggested to include the policy implications concisely at the end of this section.

3. Introduction: The novelty of the study should be improved. This section should be enriched with some graphs using statistical data.

4. Review of Literature: A separate section on literature review should be added. Sections 2 and 3 of the manuscript should be part of the review section. Furthermore, this section should be divided into two parts. The first part must be based on the theoretical foundation. The second part should be based on the empirical review directly or indirectly related to this study. Finally, find the research gap that this study fills.

5. Methodology: (1). This study was carried out for the period 1950-2010; we will soon enter 2024. It is suggested to extend the study period until 2023. (2). The flow of this section does not follow the standard format. (3). It is recommended to include a table regarding the “description of the data series” used in this study (4). It is recommended to justify the significance of each variable included in the model/estimation. (5). Referring to equation (1), in the MA part of the ARIMA model, replace t-p with t-q and also include the residual term. (6). Section 4.2 The empirical strategy should be part of the methodology section. (7). Furthermore, the use of modern econometric techniques is suggested to address the mentioned issue. (8). Please consult papers Ullah et al. (2018). Forecasting of peach area and production wise econometric analysis. Journal of Animal & Plant Sciences, 28(4); Khan et al. (2020). Forecasting area and production of guava in Pakistan: An econometric analysis. Sarhad Journal of Agriculture, 36(1), 272-281, for improving this section.

6. Results and Discussion: (1). It is recommended to revise this section in light of the above-mentioned comments (2). This section is not written in standard format. (3). Referring to Table 1, the Zivot Andrews test was not performed for FBCF. Also, include the column at last to conclude the data series as I(0)or I(1). (4). Referring to Fig. 3, it is recommended to mention the unit on both axes. (5). Correlate your study with previous studies directly or indirectly related to your study.

7. Conclusions: This section should be named “Conclusions and Policy Implications”. (1). It is suggested to Include policy implications based on study findings. (2). It is recommended to include “Future Research Directions” at the end of this section.

Reviewer #2: The theme addressed by the author may seem interesting, but its topicality and relevance are open to question. The study covers the period from 1950 to 2010. The data used is therefore not recent. The contribution to the literature is therefore questionable, insofar as the economic policy recommendations that should be based on the results found would not appear to be relevant. In addition, the overall coherence of the article is questionable, insofar as the author claims to be dealing solely with the concept of "labour productivity", but we note that he deals with the relationship between investment and productivity in a large part of his article, without it being clear why he chose this orientation.

In addition, the author makes many gratuitous statements that are not based on sources. However, when sources are cited by the author, they are very dated or poorly presented.

The central question of the study presented in the introduction, as well as the subsequent specific objectives, are not consistent with the theme dealt with.

The title of Section 2 of the article focuses on the method used and not on the key concept indicated in the title of the paper. In fact, Beveridge-Nelson decomposition is a method and not the main concept, which is the subject of the study. Consequently, this method should be discussed in the section on the methodology studied.

Furthermore, the author seems to confuse the literature review with a lesson in the history of economic thought. Indeed, the whole of section 2 refers more to a course than to a review of the literature, as is generally expected.

As for section 3, it does not seem to us to be consistent with the subject being dealt with, which is supposed to be labour productivity. In addition, no reasons are given for the period chosen, which raises the question of its relevance. Apart from the link with the subject, which is not found in this section, Figure 1, which deals with gross fixed capital formation, seems to us to be out of place. On reading this section, we find it extremely difficult to understand the real purpose of the study. Major clarifications are needed.

Section 4, and more specifically subsection 4.2 on empirical strategy, is more like a lecture than a succinct presentation of empirical strategy. In addition, the author should propose a sub-section dealing with the description of the data used and the definition of the variables mobilised.

With regard to section 5, apart from the poor numbering of the sub-sections, the author takes too little care in presenting the tables. In addition, this section still contains equations, which we find inadequate. In addition, the font used in the last paragraph of this section is not the same as that used for the other paragraphs in the section. This needs to be harmonised.

As for the conclusion, the font used is not the same as that used in the rest of the document. This needs to be harmonised.

Finally, the section on references needs to be redone in its entirety. The use of APA style is recommended.

6. PLOS authors have the option to publish the peer review history of their article (what does this mean?). If published, this will include your full peer review and any attached files.

Reviewer #1: No

Reviewer #2: No

---

## [Author Response · Author response to Decision Letter 0]

27 Aug 2024

Response to Reviewers of the Manuscript entitled “A time series analysis of labor productivity in Brazil (1950-2010)” (PONE-D-23-38514)

Dear Editor and Anonymous Reviewer, 

As the authors of the manuscript “A time series analysis of labor productivity in Brazil (1950-2010)”, we would like to thank all of you for the valuable comments and suggestions made in this round of revisions. 

Below we have the reviewers’ comments and our responses to them. We believe that we have satisfactorily addressed all the reviewers’ concerns to the best of our ability. 

This letter is to clarify how the suggestions are incorporated, as we believe they are crucial for our manuscript improvements. We reference the response to each comment, citing section, paragraph, and page number. We try to be as much specific as we can, so we highlight the modifications we have made in the original text.

Editor

After reading carefully the paper, I fully agree with the reviewers' comments regarding the novelty of the study. Why do you think this study is relevant compared to others on the same topic? The data used in the study are too old (150-2010) and all econometric tests should performed. Most of the comments raised by reviewers are common. Please carefully, respond to these comments and improve the quality of the paper.

The authors: We revised the introduction section, emphasizing the relevance on pages 1-2. The data was updated for the period from December 1991 to March 2024.

Whilst you may use any professional scientific editing service of your choice, PLOS has partnered with both American Journal Experts (AJE) and Editage to provide discounted services to PLOS authors. Both organizations have experience helping authors meet PLOS guidelines and can provide language editing, translation, manuscript formatting, and figure formatting to ensure your manuscript meets our submission guidelines. To take advantage of our partnership with AJE, visit the AJE website (http://aje.com/go/plos) for a 15% discount off AJE services. To take advantage of our partnership with Editage, visit the Editage website (www.editage.com) and enter referral code PLOSEDIT for a 15% discount off Editage services. If the PLOS editorial team finds any language issues in text that either AJE or Editage has edited, the service provider will re-edit the text for free.

Upon resubmission, please provide the following: The name of the colleague or the details of the professional service that edited your manuscript. A copy of your manuscript showing your changes by either highlighting them or using track changes (uploaded as a *supporting information* file). A clean copy of the edited manuscript (uploaded as the new *manuscript* file)

The authors: This version of the article has undergone extensive grammatical review by the authors. The authors: We submitted our version of the manuscript highlighting the revisions made.

3. In the online submission form, you indicated that Data will be shared upon request.

All PLOS journals now require all data underlying the findings described in their manuscript to be freely available to other researchers, either a. In a public repository, b. Within the manuscript itself, or c. Uploaded as supplementary information. This policy applies to all data except where public deposition would breach compliance with the protocol approved by your research ethics board. If your data cannot be made publicly available for ethical or legal reasons (e.g., public availability would compromise patient privacy), please explain your reasons on resubmission and your exemption request will be escalated for approval. 

The authors: We now send the data used to estimate the model as supplementary information.

The authors: We include the following statement at the end of the manuscript, before the references:

Data availability statement: The data used to estimate the model are freely available online and can be accessed through the following links: http://www.ipeadata.gov.br/Default.aspx. The data in the Excel spreadsheet is sent to the Plos One Editors along with this manuscript.

Reviewers' comments:

Reviewer's Responses to Questions

Comments to the Author

1. Is the manuscript technically sound, and do the data support the conclusions?

Reviewer #1: Yes

Reviewer #2: Partly

2. Has the statistical analysis been performed appropriately and rigorously?

Reviewer #1: No

Reviewer #2: No

3. Have the authors made all data underlying the findings in their manuscript fully available?

Reviewer #1: No

Reviewer #2: No

4. Is the manuscript presented in an intelligible fashion and written in standard English?

Reviewer #1: Yes

Reviewer #2: Yes

5. Review Comments to the Author

Reviewer #1:

1. Title: The title of this study is general and should be more specific. The study period should not be mentioned in the title.

The authors: We revised the paper title to Trade liberalization and total factor productivity in Brazil: a VECM modeling.

2. Abstract: (1). The results of the study are in a qualitative way. It is suggested to report here the findings of the study in a quantitative way (2). It is suggested to include the policy implications concisely at the end of this section.

The authors: The abstract has also been revised, now including some quantitative results.

3. Introduction: The novelty of the study should be improved. This section should be enriched with some graphs using statistical data.

The authors: In this revised version, we summarize the literature review rationally. We begin Section 1 with more general questions and then justify why this work is relevant (on page 1). Next, we summarize existing studies and argue for our contribution to this literature (on pages 2-3). We also justify the relevance of this study to the Brazilian case, which is reinforced by Figure 1.

4. Review of Literature: A separate section on literature review should be added. Sections 2 and 3 of the manuscript should be part of the review section. Furthermore, this section should be divided into two parts. The first part must be based on the theoretical foundation. The second part should be based on the empirical review directly or indirectly related to this study. Finally, find the research gap that this study fills.

The Authors: This version includes a literature review section (Section 2). The first part of this section focuses on some theoretical aspects. Then, we discuss the literature's findings. Finally, we present a descriptive discussion of the Brazilian case.

5. Methodology: (1). This study was carried out for the period 1950-2010; we will soon enter 2024. It is suggested to extend the study period until 2023. (2). The flow of this section does not follow the standard format. (3). It is recommended to include a table regarding the “description of the data series” used in this study (4). It is recommended to justify the significance of each variable included in the model/estimation. (5). Referring to equation (1), in the MA part of the ARIMA model, replace t-p with t-q and also include the residual term. (6). Section 4.2 The empirical strategy should be part of the methodology section. (7). Furthermore, the use of modern econometric techniques is suggested to address the mentioned issue. (8). Please consult papers Ullah et al. (2018). Forecasting of peach area and production wise econometric analysis. Journal of Animal & Plant Sciences, 28(4); Khan et al. (2020). Forecasting area and production of guava in Pakistan: An econometric analysis. Sarhad Journal of Agriculture, 36(1), 272-281, for improving this section.

The Authors: We have rewritten the methodology section and cited the suggested articles to strengthen the choice of the VECM approach (penultimate paragraph of subsection 3.1 on page 7).

6. Results and Discussion: (1). It is recommended to revise this section in light of the above-mentioned comments (2). This section is not written in standard format. (3). Referring to Table 1, the Zivot Andrews test was not performed for FBCF. Also, include the column at last to conclude the data series as I(0)or I(1). (4). Referring to Fig. 3, it is recommended to mention the unit on both axes. (5). Correlate your study with previous studies directly or indirectly related to your study.

The authors: This section has been rewritten. We begin with the econometric tests and then discuss the model's results and its consistency with the findings in the literature and with Brazilian reality.

7. Conclusions: This section should be named “Conclusions and Policy Implications”. (1). It is suggested to Include policy implications based on study findings. (2). It is recommended to include “Future Research Directions” at the end of this section.

The authors: We also review the study's findings. In this section, we provide policy implications and specific recommendations, and future research agenda. 

Reviewer #2:

The theme addressed by the author may seem interesting, but its topicality and relevance are open to question. The study covers the period from 1950 to 2010. The data used is therefore not recent. The contribution to the literature is therefore questionable, insofar as the economic policy recommendations that should be based on the results found would not appear to be relevant. In addition, the overall coherence of the article is questionable, insofar as the author claims to be dealing solely with the concept of "labour productivity", but we note that he deals with the relationship between investment and productivity in a large part of his article, without it being clear why he chose this orientation. 

In addition, the author makes many gratuitous statements that are not based on sources. However, when sources are cited by the author, they are very dated or poorly presented. The central question of the study presented in the introduction, as well as the subsequent specific objectives, are not consistent with the theme dealt with.

The authors: We have addressed the concerns raised by the reviewer in this revised version of the manuscript. We have completely revised this article, updating the references. First, we update the data from December 1991 to March 2024. Now, we draw on a vast international trade literature that associates productivity with trade openness. Some studies also argue that absorptive capacity explains productivity. Based on these studies, we specify the VECM econometric model and justify the choice of this model over other existing forecasting methodologies, such as the BOX-Jenkins methodology (section 3). Next, we perform diagnostic and specification statistical tests. Finally, we discuss the results and their consistency with past findings and with Brazil's specific case (Section 4). We conclude by presenting the policy implications of this study and suggestions for formulating a policy that aims to enhance the effect of trade openness on productivity.

The title of Section 2 of the article focuses on the method used and not on the key concept indicated in the title of the paper. In fact, Beveridge-Nelson decomposition is a method and not the main concept, which is the subject of the study. Consequently, this method should be discussed in the section on the methodology studied. Furthermore, the author seems to confuse the literature review with a lesson in the history of economic thought. Indeed, the whole of section 2 refers more to a course than to a review of the literature, as is generally expected.

The authors: We update the data and present the rationale for the choice of model variables and the VECM method. Thus, the Beveridge-Nelson decomposition is only an empirical strategy to separate the short—and long-term components of TFP.

As for section 3, it does not seem to us to be consistent with the subject being dealt with, which is supposed to be labour productivity. In addition, no reasons are given for the period chosen, which raises the question of its relevance. Apart from the link with the subject, which is not found in this section, Figure 1, which deals with gross fixed capital formation, seems to us to be out of place. On reading this section, we find it extremely difficult to understand the real purpose of the study. Major clarifications are needed.

The authors: Revisions made are highlighted throughout the text. Regarding the purpose of the study, we review section 1, where we present the study's objective, motivation, and relevance. We conclude this section by showing the study's contribution to the existing literature.

Section 4, and more specifically subsection 4.2 on empirical strategy, is more like a lecture than a succinct presentation of empirical strategy. In addition, the author should propose a sub-section dealing with the description of the data used and the definition of the variables mobilised.

The authors: We have revised the methodology section. We now present the model's theoretical aspects and conclude with the decomposition of productivity (section 3).

With regard to section 5, apart from the poor numbering of the sub-sections, the author takes too little care in presenting the tables. In addition, this section still contains equations, which we find inadequate. In addition, the font used in the last paragraph of this section is not the same as that used for the other paragraphs in the section. This needs to be harmonised.

The authors: We eliminated the exaggerated subdivisions of the sections and formatted and standardized the tables.

As for the conclusion, the font used is not the same as that used in the rest of the document. This needs to be harmonised.

The authors: We reviewed the sections of the manuscript version. The sections were harmonized.

Finally, the section on references needs to be redone in its entirety. The use of APA style is recommended.

The authors: We

---

## [Decision Letter · Decision Letter 1]

10 Oct 2024

Trade liberalization and total factor productivity in Brazil: a VECM modeling

PONE-D-23-38514R1

Dear Julio Vicente Cateia,

We’re pleased to inform you that your manuscript has been judged scientifically suitable for publication and will be formally accepted for publication once it meets all outstanding technical requirements.

Kind regards,

Opeoluwa Adeniyi Adeosun

Academic Editor

PLOS ONE

**Comments to the Author**

1. All comments have been addressed

2. Is the manuscript technically sound, and do the data support the conclusions?

Yes

3. Has the statistical analysis been performed appropriately and rigorously? 

Yes

4. Have the authors made all data underlying the findings in their manuscript fully available?

Yes

5. Is the manuscript presented in an intelligible fashion and written in standard English?

Yes

6. Review Comments to the Author

The comments have been largely addressed. The authors made significant efforts to update the data and rerun the model. They also reinterpreted the analysis and added more value to the paper.

---

## [Editor Report · Acceptance letter]

30 Oct 2024

PONE-D-23-38514R1 

PLOS ONE

Dear Dr. Cateia, 

I'm pleased to inform you that your manuscript has been deemed suitable for publication in PLOS ONE. Congratulations! Your manuscript is now being handed over to our production team.

Kind regards, 

on behalf of

Dr. Opeoluwa Adeniyi Adeosun 

Academic Editor

PLOS ONE